# MicroRNA-Mediated Regulation of Histone-Modifying Enzymes in Cancer: Mechanisms and Therapeutic Implications

**DOI:** 10.3390/biom13111590

**Published:** 2023-10-28

**Authors:** Joanna Szczepanek, Andrzej Tretyn

**Affiliations:** 1Centre for Modern Interdisciplinary Technologies, Nicolaus Copernicus University, ul. Wilenska 4, 87-100 Torun, Poland; 2Faculty of Biological and Veterinary Sciences, Nicolaus Copernicus University, ul. Lwowska 1, 87-100 Torun, Poland; prat@umk.pl

**Keywords:** microRNA, histone-modifying enzymes, cancer, gene regulation, epigenetics, therapeutic implications

## Abstract

In the past decade, significant advances in molecular research have provided a deeper understanding of the intricate regulatory mechanisms involved in carcinogenesis. MicroRNAs, short non-coding RNA sequences, exert substantial influence on gene expression by repressing translation or inducing mRNA degradation. In the context of cancer, miRNA dysregulation is prevalent and closely associated with various stages of carcinogenesis, including initiation, progression, and metastasis. One crucial aspect of the cancer phenotype is the activity of histone-modifying enzymes that govern chromatin accessibility for transcription factors, thus impacting gene expression. Recent studies have revealed that miRNAs play a significant role in modulating these histone-modifying enzymes, leading to significant implications for genes related to proliferation, differentiation, and apoptosis in cancer cells. This article provides an overview of current research on the mechanisms by which miRNAs regulate the activity of histone-modifying enzymes in the context of cancer. Both direct and indirect mechanisms through which miRNAs influence enzyme expression are discussed. Additionally, potential therapeutic implications arising from miRNA manipulation to selectively impact histone-modifying enzyme activity are presented. The insights from this analysis hold significant therapeutic promise, suggesting the utility of miRNAs as tools for the precise regulation of chromatin-related processes and gene expression. A contemporary focus on molecular regulatory mechanisms opens therapeutic pathways that can effectively influence the control of tumor cell growth and dissemination.

## 1. Introduction

MicroRNAs (miRNAs) represent a class of short, single-stranded non-coding RNA molecules, typically comprising 18–25 nucleotides, that play intricate regulatory roles in post-transcriptional gene expression. These small RNAs are most important in numerous biological processes, including development, differentiation, and disease [1,2]. Of particular interest is their involvement in cancer, where they have emerged as key players in coordinating the complex landscape of oncogenic and tumor-suppressive pathways [3,4,5].

MiRNAs function by binding to specific target mRNAs through partial complementarity, primarily within the 3′ untranslated region (UTR), resulting in translational repression or degradation of the target transcript [6,7,8]. Some miRNAs act as oncogenes, promoting tumor growth, while others function as tumor suppressors, inhibiting cellular transformation and metastasis [9,10,11,12] (Table 1). These differential roles have prompted researchers to investigate miRNAs as potential diagnostic and prognostic biomarkers.

The process of cancer development is multi-stage, and genetic changes occurring at each stage lead to significant dysregulation of proteins involved in the regulation of the cell cycle. There is a hypothesis that these changes may be the result of miRNA interaction [103,104,105]. MiRNAs have the ability not only to directly block the expression of specific target genes, but also influence the regulation of the expression of epigenetic modifiers, such as histone-modifying enzymes. They play an important role in causing structural changes in chromosomes [106,107,108,109]. Hence, microRNAs may participate in the regulation of epigenetic mechanisms in the context of cancer, which often leads to inappropriate expression profiles in malignant tumors.

Concurrently, histone-modifying enzymes have received significant attention due to their important role in regulating gene expression through epigenetic modifications. Chromatin, a dynamic complex of DNA and histone proteins, is subject to various modifications, such as acetylation, methylation, and phosphorylation, which influence chromatin structure and accessibility to the transcriptional machinery [110,111]. Aberrant histone modifications have been implicated in the initiation and progression of various cancers by disrupting gene expression patterns and perturbing normal cellular processes [110,111,112].

This manuscript delves into the intricate interplay between miRNAs and histone-modifying enzymes in the context of cancer. It explores the diverse mechanisms through which miRNAs modulate the activity of these enzymes, subsequently affecting chromatin structure and gene expression. Additionally, we examine the consequential impact on critical cellular processes such as proliferation, differentiation, and apoptosis.

As our understanding of these regulatory mechanisms becomes more comprehensive, emerging therapeutic possibilities become apparent. Directing the interaction between miRNA and histones exhibits potential in devising ground-breaking strategies to address malignancy. Through the explication of miRNAs’ roles in governing epigenetic alterations, we endeavor to pave the pathway for enhanced precision and effectiveness in therapeutic interventions for cancer treatment.

## 2. Histone-Modifying Enzymes in Chromatin Regulation

Histones are lysine- and arginine-rich proteins involved in chromosome condensation, consisting of four core types (H2A, H2B, H3, and H4) located in the nucleosome bead, along with two linker histones (H1/H5). The amino- and carboxyl- termini of these proteins may be modified [113,114,115,116] (Figure 1).

Epigenetic control of gene expression is a complex and tightly regulated process crucial for normal cellular functions. One of the key mechanisms underlying epigenetic regulation is histone modification, which involves the covalent alteration of histone proteins within chromatin [106,110,117]. Histone-modifying enzymes play a central role in adding, removing, or interpreting these modifications, thereby influencing chromatin structure and gene expression (Table 2 and Table 3). Depending on the specific histone marker and reader protein, this interaction can lead to gene activation or repression. Histone modifications are dynamically regulated by a set of enzymes known as “writers”, “erasers”, and “readers”. Writers, such as histone acetyltransferases (HATs) and histone methyltransferases (HMTs), add specific marks to histones. Erasers, including histone deacetylases (HDACs) and histone demethylases, remove these marks. Readers, such as bromodomain-containing proteins for acetylation and chromodomains for methylation, interpret the modified histones and recruit other components of the epigenetic machinery [118,119,120]. These enzymes collectively manage the dynamic histone modification landscape that governs gene expression (Table 2 and Table 3). HATs catalyze the addition of acetyl groups to lysine residues on histone tails. This modification, known as histone acetylation, generally relaxes chromatin structure, allowing the increased accessibility of transcriptional machinery to gene promoters. HATs are critical for activating gene expression and are often associated with euchromatin regions [121,122,123]. HDACs remove acetyl groups from histone lysine residues, leading to chromatin condensation and transcriptional repression. HDACs are associated with gene silencing and are often found in heterochromatin regions [124,125,126,127]. HMTs catalyze the addition of methyl groups to specific lysine or arginine residues on histone tails. Histone methylation can lead to either transcriptional activation or repression, depending on the target residue and the degree of methylation. HMTs play a key role in maintaining gene expression patterns and cellular identity [111,127,128]. Histone demethylases remove methyl groups from histone lysine or arginine residues, contributing to the dynamic regulation of gene expression. The removal of methyl marks can either activate or repress gene transcription, depending on the specific residue and context [129,130]. Bromodomain-containing proteins specifically recognize acetylated lysine residues on histones and recruit various transcriptional regulators to promote gene expression. Bromodomain readers are involved in the assembly of transcriptional complexes and chromatin remodeling [131,132,133]. Chromodomain readers bind to methylated histone residues and play a role in mediating gene expression by recruiting chromatin-modifying complexes [134,135].

Dysregulation of histone-modifying enzymes has been strongly implicated in cancer development and progression (Figure 1). Aberrant activity or expression of these enzymes can lead to altered chromatin states, resulting in disrupted gene expression profiles and genomic instability [60,107,111,112,117,201,202,203,204,205]. Some key points of implication include:(1)Oncogene Activation and Tumor Suppressor Silencing: Histone acetylation and methylation are often associated with gene regulation, influencing the activation or repression of genes, including oncogenes that drive tumorigenesis [112,205]. Mutations or overexpression of HATs and HMTs can lead to the hyperactivation of oncogenes, contributing to uncontrolled cell growth [138,143,206]. Conversely, the silencing of tumor suppressor genes through histone deacetylation and methylation is a hallmark of many cancers. HDACs and histone demethylases can contribute to the epigenetic silencing of genes that regulate cell cycle control and DNA repair [143,201,207].(2)Epigenetic Plasticity and Drug Resistance: Cancer cells often exhibit epigenetic plasticity, allowing them to adapt to changing environments and develop resistance to therapies [208]. Histone-modifying enzymes contribute to this plasticity by maintaining specific chromatin states that promote drug resistance, making them attractive targets for therapeutic intervention [208,209,210].(3)Diagnostic and Therapeutic Targets: Aberrant histone modifications and their associated enzymes can serve as potential diagnostic markers for certain cancer types. Furthermore, targeting histone-modifying enzymes with small molecule inhibitors holds promise as a therapeutic strategy to restore normal gene expression patterns in cancer cells [201,211,212].

## 3. MicroRNA-Mediated Regulation of Histone Modifications

Studies on the regulation of gene expression have shown that, in addition to classical transcription mechanisms, there are also subtle but important ways of controlling gene activity at the epigenetic level. One of the key aspects of epigenetics is histone modification, whose dynamic changes affect the availability of chromatin for transcription complexes and thus gene expression. In this context, microRNAs have the ability to interact with epigenetic factors, including histone-modifying enzymes [106,107,213] (Figure 2). Traditionally known for their post-transcriptional gene silencing effects, recent research has highlighted a fascinating dimension of miRNA function–their direct interactions with histone-modifying enzymes. These interactions contribute to the intricate web of epigenetic control, influencing chromatin structure, gene expression patterns, and, ultimately, cellular behavior [106,214,215,216]. In the context of oncology, disruptions in miRNA–histone modifier interactions have significant clinical implications, contributing to tumor development and progression (Figure 2). There are two main types of interactions between miRNAs and histone-modifying enzymes: direct interactions of microRNAs with enzymes and indirect mechanisms linking miRNAs with chromatin remodeling [217,218].

The interaction between miRNAs and histone-modifying enzymes involves direct binding to mRNA, leading to altered expression of histone modifiers. Many miRNAs can also directly interact with proteins regulating epigenetic states through histone modifications. Examples of such miRNAs include: miR-29a (MYC/HDAC3/EZH2 in lymphoma [219]), miR-200a (HDAC4/SP1 in hepatocellular carcinoma [220]), miR-224 (HDAC1/HDAC3/EP300 in hepatocellular carcinoma [221]), miR-212 (EZH2/G9a/HDAC in lung cancer [222]), miR-126 (HDAC2 in prostate cancer [223]), miR-34 a (SIRT1 in breast cancer [224,225]), miR-34b (HDAC1/HDAC2/HDAC4 in prostate cancer [226]), miR-127, -411, -431 and -432 (HDACs in osteosarcoma [227], miR-9-5p (HDACs in gastric cancer [228]), mir-101 (EZH2 in glioblastoma [229]), miR-22 (TIP60 and HDAC4 in breast cancer [230,231]), miR-34a (HDAC1 in ovarian cancer [232]), miR-125 (HDAC4,5 in breast cancer [233,234]), miR-142 (ASH1L (KMT2H) in leukemia and thyroid cancer [235,236]), miR-675 (SUV39H2 (KMT1B) in liver cancer [237], miR-122 (SUV39H1 (KMT1A) in HCC [238]), miR-101 (KMT6 (EZH2) in NSCLC, prostate and renal cancer [239,240,241]), miR-195 (PRMT4 (CARM1) in colorectal cancer [242]), or miR-155 (JMJD1A in nasopharyngeal carcinoma [243]. For instance, miRNAs can target HATs and HDACs, influencing the acetylation status of histones and affecting transcriptional activation or repression [218,244]. Similarly, miRNAs can regulate HMTs and histone demethylases, leading to changes in histone methylation patterns associated with gene expression changes [204,245]. The interaction between miRNA and HDAC has been well-documented both in human cancers tissues and in cancer cell lines. For example, in the prostate cancer cell line, the downregulation of miR-101, miR-449a and miR-17-5p has been observed to result in the upregulation of their respective targets: EZH2 protein, HDAC-1, and p300/CBP- associated factor. This, in turn, leads to the stimulation of cell growth and the progression of prostate cancer [246,247]. One important example of direct miRNA interactions with histone-modifying enzymes is related to miRNA-449a and histone deacetylase 1. MiR-449a is downregulated in prostate cancer tissues compared to controls, suggesting a potential tumor-suppressing role. Introducing miR-449a into prostate cancer cells induces cell cycle arrest, apoptosis, and a senescence-like phenotype, indicating its impact on cancer cell behavior. Analysis of the 3′-UTR regions identifies HDAC-1, overexpressed in cancer, as a direct target of miR-449a. Studies demonstrate that miR-449a inhibits HDAC-1 expression, suggesting that, through this regulation, the microRNA can influence the growth and survival processes of prostate cancer cells [246].

In cancer, many miRNAs are regulated by histone methylation, creating feedback loops between miRNAs and methylation pathways. Increased expression of miRNAs such as miR-101, miR-125a-5p, miR-122, miR-675, miR-212, miR-22-3p, miR-142, and miR-181a can affect histone methyltransferases and subsequently influence chromatin structure [67,222,235,236,237,238,248,249,250,251,252,253,254,255,256,257,258,259]. MiRNAs also influence histone demethylation. For instance, miR-133b regulates the expression levels of *DOT1L*, while miR-502 and miR-7 interact with the SET8 enzyme [253,254,256,257,258]. MiRNAs such as miR-101, miR-26a, miR-137, miR-124, miR-138, miR-31, and miR-98, impact the expression of enzymes involved in histone methylation regulation, such as EZH1, EZH2, and G9a [260,261,262,263,264,265,266,267,268,269,270]. EZH1 and EZH2 enzymes regulate the histone H3 lysine 27 methylation. MiR-17-5p reduction elevates KMT6B (EZH1), causing erlotinib resistance in NSCLC [259]. MiR-93, a member of the mir106b-25 cluster, plays a dual role in oncogenesis. While often overexpressed in human malignancies, it can also function as a tumor suppressor. Downregulation relates to aggressive breast cancer [271]. It regulates stem cell regulatory genes, such as *JAK1*, *STAT3*, *AKT3*, *SOX4*, *EZH1*, and *HMGA2*, affecting the fate of both normal and malignant mammary stem cells [272]. MiR-101’s impact on EZH2 is evident across cancers, like NSCLC, prostate, and renal cancers [260,261,264,269]. MiRNA-101 affects histone modifications by regulating the expression of two histone-modifying enzymes: EZH2 (PRC2 methyltransferase enzyme) and DOT1L (H3K79 methyltransferase enzyme) [269,273,274,275]. In some cancers, such as prostate cancer, decreased miRNA-101 expression is observed, leading to increased EZH2 expression [240]. MiR-26a, an apoptosis inducer, regulates EZH2; decreased miR-26a occurs in lung, rhabdomyosarcoma, and prostate cancers [265,266,276]. MiR-26a prevents Burkitt lymphoma and prostate cancer via c-myc inhibition [276,277]. In hepatocellular carcinoma, miR-137 reduction is associated with EZH2 elevation, inhibiting progression. MiR-124 similarly affects EZH2 in cancer cell lines [67]. MiR-138 deficiency drives metastasis via VIM, ZEB2, and EZH2 targeting [278]. The impact of miR-98 on human ovarian cancer stem cells (OCSCs) is evident as p21(CIPI/WAF1) saw an increase in expression, while the CDK2/cyclin E complex and c-Myc were downregulated. Notably, changes were observed in the levels of E2F1, retinoblastoma protein (pRb), and HDAC1 within the pRb-E2F signaling pathway. Most significantly, miR-98 suppressed the growth of OCSCs’ xenograft tumors. These findings highlight miR-98’s potential to effectively inhibit in vitro cell proliferation and modulate the pRb-E2F pathway in human OCSCs [268]. MiRNA-29a is another example of a microRNA that directly interacts with a histone-modifying enzyme. In this case, miRNA-29a regulates the expression of HMT, which introduces methyl groups on histones. MiRNA-29a is known to regulate processes related to DNA and histone methylation [55,279]. Reduced miRNA-29a expression is often observed in lung cancer, leading to increased DNMT3A activity. Increased histone methylation may result in changes in chromatin structure and abnormal expression of tumor suppressor genes, contributing to carcinogenesis. miR-16 can inhibit the expression of HDAC4 (histone deacetylase 4), which leads to the accumulation of active acetylated histone H4. This, in turn, leads to the inhibition of the cell cycle and a reduction in cancer cell proliferation. MiR-34a is an example of a miRNA that is induced by p53 and can inhibit the expression of the SIRT1 (histone deacetylase 1) gene, leading to the accumulation of active acetylated histone H4 and increased apoptosis (Table 4).

The indirect mechanisms linking miRNAs to chromatin remodeling are intricate networks of interactions that influence the epigenetic landscape and gene expression patterns. These mechanisms do not involve direct interactions between miRNAs and histone-modifying enzymes. Instead, they operate through intermediary factors that regulate the chromatin remodeling processes. The following outlines these mechanisms:(1)Transcription Factors and Transcriptional Regulation: miRNAs can indirectly regulate chromatin remodeling by targeting transcription factors (TFs) or co-regulators involved in chromatin modification [107,313]. When miRNAs target TFs controlling the expression of histone-modifying enzymes, downstream changes in histone modifications occur [107,313,314,315]. MiRNAs, like miRNA-200, indirectly influence histone modifications through interactions with transcription factors. In this case, miRNA-200 interacts with ZEB1 and ZEB2, repressors of E-cadherin, impacting cell adhesion and promoting metastasis [316]. Another example is miR-29b, repressed by the MYC protein in KIT-mutation-associated leukemia. This leads to increased *Sp1* expression, which activates *KIT* gene transcription. Synthetic miR-29b inhibitors disrupt this network, reducing KIT expression and inhibiting leukemia growth [317].(2)Signaling Pathways and Epigenetic Modulators: Signaling pathways, including Wnt, Notch, and TGF-β, are modulated, which in turn affects epigenetic regulators [12,318,319]. For instance, miR-29 and miR-206 impact the TGF-β pathway and HDAC4 expression, crucial for myogenic genes. Reduced miR-29 and miR-206 levels lead to increased HDAC4 expression by inhibiting its translation. They also regulate the Smad3 levels, a key TGF-β pathway component, impacting muscle cell differentiation. MiR-29 and miR-206 counteract TGF-β’s negative effects on cell commitment, and their overexpression inhibits rhabdomyosarcoma development [320,321]. Interestingly, rhabdomyosarcoma tumors exhibit elevated *TGF-β* and Smad4, coinciding with our findings that increased TGF-β signaling suppresses these miRNAs, affecting cellular differentiation [320,321,322,323,324].(3)Long Non-Coding RNAs (lncRNAs) and RNA Interference: LncRNAs act as intermediaries between miRNAs and chromatin remodeling [218,325]. When an miRNA represses an lncRNA, it increases the expression of genes targeted by the lncRNA. LncRNAs interact with chromatin modifiers, indirectly affecting histone modifications [106,218,326]. This intricate regulatory network connects miRNAs and chromatin remodeling, as some miRNAs and lncRNAs share target genes. For example, the lncRNA HOTAIR plays a significant role in cancer progression by affecting prognosis, staging, and multiple cellular processes through miRNA modulation. HOTAIR primarily influences chromatin remodeling and epigenetic changes by acting as a scaffold for histone-modifying protein complexes. It facilitates gene silencing through H3K27 methylation and H3K4 demethylation and is known to promote metastasis by epigenetically silencing the tumor suppressor gene miR-34a. Various miRNAs, including miR-7, miR-206, miR-218, miR-20a-5p, miR-126-5p, and miR-146a-5p, are involved in regulating HOTAIR’s effects [327,328,329,330].(4)DNA Methylation and Epigenetic Crosstalk: DNA methylation, closely linked to histone modifications, is indirectly influenced by miRNAs targeting DNMTs or DNA demethylation factors. These changes in DNA methylation impact chromatin structure, thereby altering histone modifications and gene expression [331,332,333]. For example, miR-101 inhibits *DNMT3A* expression, leading to increased DNA methylation in tumor-suppressing gene promoters, affecting chromatin accessibility for histone-modifying enzymes. This results in altered histone modifications, ultimately influencing gene expression. MiR-101 also reverses *PRDM16* gene promoter hypomethylation by modifying histones, which are mediated through direct targets such as *EZH2*, *EED*, and *DNMT3A*, suggesting their role in cancer contexts [334].(5)Chromatin Remodeling Complexes: miRNAs indirectly regulate chromatin remodeling by targeting complex components, impacting chromatin structure and access to DNA. For instance, miR-124 and miR-9 inhibit BAF complex (SWI/SNF) activity by reducing BAF subunit expression [335]. The BAF complex plays a key role in chromatin remodeling and gene expression regulation [334]. Reduced microRNA levels lead to increased *BAF* expression, affecting complex activity, and ultimately influencing gene expression through chromatin structure [335].

## 4. Functional Consequences of miRNA-Histone Enzyme Interplay

The interactions between miRNAs and histone-modifying enzymes have a significant impact on many cellular processes, such as proliferation, differentiation, and apoptosis. In the context of oncology, these interactions play a key role in regulating the expression of genes related to processes such as disease initiation and progression, metastasis, angiogenesis, and resistance to therapy in numerous cancers (Figure 1).

First of all, the interactions between miRNAs and histone-modifying enzymes are an important mechanism for regulating cell proliferation. MiRNAs such as miR-125b, the miR-17-92 cluster, and miR-34a have been identified as regulators of proliferation processes in breast cancer [336,337,338]. MiR-125b can influence HDAC expression levels, which has an impact on histone activation and the regulation of the expression of key cell cycle genes [233,234,339]. The miR-17-92 cluster interacts with HDAC6, which affects the expression of cyclin D1 and CDK4-related genes, stimulating proliferation [340,341]. However, miR-34a, which is activated by p53, inhibits the expression of *CYCLIN E2* and *CDK6*, which leads to a block in the G1 phase of the cell cycle [342,343,344,345]. The conducted research revealed that miRNA inhibited the acetylation of p53 by interacting with both histone deacetylase 1 (HDAC1) and the E1A binding protein p300. This interaction led to the suppression of p53 activity, consequently promoting tumor growth and resistance to chemotherapy. In solid cancer and hematological malignancies, miRNAs such as miR-15a/miR-16-1 and miR-29b can influence the expression of histone enzymes and regulate proliferation processes [56,346,347,348]. MiR-15a/miR-16-1 can interact with HDAC3 and HDAC9, which affects histone activity and the expression of cell cycle-related genes [346,348]. However, miR-29b interacts with DNMT3A and DNMT3B, affecting DNA methylation and the expression of key genes used in the cell cycle [317,346,349]. In colorectal cancer, miRNAs such as miR-449a have been implicated in cell cycle regulation through their interactions with histone-modifying enzymes. MiR-449a inhibits the expression of HDAC1 and HDAC4, leading to the accumulation of active acetylated histones and the increased expression of cell cycle regulatory genes such as *CDKN1A* [350,351,352].

Regulation of miRNA interactions with histone-modifying enzymes is also crucial for cell differentiation processes. MiRNAs can interact with enzymes that modulate histone modifications, affecting chromatin structure and accessibility of transcription sites [106,108,313]. For example, miR-221 and miR-222, which are often overexpressed in tumors, inhibit the expression of KDM6A (UTX), an enzyme that demethylates histone H3K27. The decrease in KDM6A activity leads to the maintenance of strong methyl groups on histone H3K27, which in turn affects the maintenance of condensed chromatin in gene regions responsible for stem cell differentiation. This may result in the inhibition of the expression of differentiation genes and promote the maintenance of cells in an undifferentiated state [353,354,355]. In breast cancer, miR-486 overexpression has been associated with maintaining cells in an undifferentiated state. MiR-486 can influence cell differentiation by interacting with the JARID1B (KDM5B) enzyme, which is responsible for removing the methyl group from histone H3K4 [356]. Low JARID1B activity leads to the accumulation of methylation on H3K4, which hinders the activation of differentiation genes, contributing to the maintenance of cells in an undifferentiated state [356,357,358,359]. A similar effect may be observed in HPSCC, lung cancer, bladder cancer, colorectal cancer, prostate cancer, and malignant melanoma [359]. In numerous cancers, the miR-200 family of miRNAs plays a key role in regulating differentiation. MiR-200c can affect the EZH2 enzyme, which is responsible for the methylation of histone H3K27me3 in gene areas responsible for the regulation of cellular differentiation. In prostate cancer, miR-200c plays a key role as a mediator between the *EZH2* and the *E2F3* genes. MiR-200c regulates *E2F3* by directly binding to the 3’UTR region of the *E2F3* gene. Downregulation of miR-200c reduces the impact of *EZH2* depletion on *E2F3* expression and cell cycle processes. As a result, this mechanism creates a feedback loop that is crucial for cancer development [360]. Another mechanism of indirect interaction, through lncRNA, is also possible. The lncRNA *SNHG22* contributes to gastric cancer progression through a complex regulatory mechanism. Its role consists of two main aspects, in which miR-200c-3p plays a key role. SNHG22 increases the expression of the Notch1 oncogene by “covering” this miRNA while recruiting EZH2, which leads to the silencing of the tumor suppressor genes [361]. In gastric cancer, SNHG22 recruits EZH2 to influence the level of chromatin condensation, particularly in the promoters of many tumor suppressor genes, such as *E-cadherin*, *EAF2*, *ADRB2*, *rap1GAP*, and *RUNX3*. It has been shown that miR-627 is responsible for reducing the expression of the *JMJD1A* gene, which encodes a histone demethylase. By downregulating JMJD1A, miR-627 increases histone H3K9 methylation and inhibits the expression of proliferative factors such as GDF15. Overexpression of miR-627 inhibited the proliferation of CRC cell lines in culture and the growth of xenogeneic tumors in mice [362].

Moreover, miRNA interactions with histone-modifying enzymes play a key role in regulating the apoptosis process. MiRNAs can influence the activity of histone enzymes that control the availability of transcription sites associated with apoptosis. For example, miR-125b, which is often associated with carcinogenesis, can regulate the expression of the enzyme EZH2, responsible for the methylation of histone H3K27. MiR-125b inhibits EZH2 expression, which leads to decreased histone H3K27 methylation in the regulatory regions of apoptotic genes. This results in the activation of the expression of these genes and the promotion of apoptotic pathways. In breast cancer, miR-29b is often downregulated, which affects the regulation of apoptosis through interaction with the DNMT3A enzyme. This miRNA inhibits *DNMT3A* expression, potentially contributing to altered expression of apoptotic genes (*BCL2* and *MCL1*). DNMT3A, an enzyme responsible for DNA methylation, can modulate the expression of genes key to the induction of apoptosis.

In ovarian cancer, miR-101 is often downregulated, leading to the evasion of apoptosis. MiR-101 interacts with the EZH2 enzyme, known for its involvement in carcinogenesis. The mechanism by which it does this is the inhibition of *EZH2* expression by miR-101, which results in a decrease in the histone H3K27 methylation in the areas regulating genes associated with apoptosis. A decrease in miR-101 expression may lead to a decrease in *FAS* and *BIM* expression, which in turn may reduce the ability of cells to induce apoptosis. In leukemias, miR-15a/miR-16-1 is often overexpressed, affecting the expression of genes related to apoptosis, such as the *BCL2* and *BCL2L2*-anti-apoptotic genes from the *BCL2* family. Overexpression of miRNAs may lead to a decrease in the expression of these genes, which may contribute to an increase in the propensity of cells to undergo apoptosis. In lung cancer, miR-34a, acting as an apoptosis-regulating factor, is often downregulated in various types of cancer. MiR-34a affects histone-modifying enzymes, such as HDAC1 and SIRT1, which affect histone activation. The effect is a decrease in the expression of anti-apoptotic genes, such as *BCL2*, *SIRT1*, and *NOTCH1*, which may promote increased apoptosis induction. In in vivo and in vitro cancer, HDAC inhibitors reduced breast cell tumorigenesis via the activation of intrinsic apoptosis through the caspase 9/3 pathway. This process involves the HDACi-mediated expression of miR-125a-5p, which is achieved by activating the RUNX3/p300/HDAC5 complex. Consequently, miR-125a-5p post-transcriptionally silenced HDAC5, creating a regulatory loop in breast cancer cells, controlled by RUNX3 signaling, that inhibited cancer progression and promoted apoptosis when miR-125a-5p and RUNX3 were active, but had the opposite effect when HDAC5 was silenced [234].

The progression and prognosis of cancer are significantly influenced by processes related to angiogenesis, resulting from hypoxia induced by competition between rapidly dividing cancer cells [363,364,365]. The regulation of these angiogenesis processes in the context of cancer development is closely related to the activity of histone-modifying enzymes and their control by miRNA. One of the key participants in this mechanism is EZH2, which has the ability to induce angiogenesis and promote tumor growth [366]. miR-137 is an important element in the regulation of this process by affecting the transcript of the EZH2 enzyme. MiRNA-137, being a tumor suppressor, binds to the 3′-UTR of this gene, leading to a reduction in its expression. This interaction between miRNA-137 and EZH2 contributes to limiting the proliferation of cancer cells and the angiogenesis process. Such an interaction has been described in the case of a glioblastoma [367]. A similar effect may be observed in the case of changes in miR-101 expression profiles [229]. In the context of reduced expression of tumor suppressors, miRNA plays an important role in the activation of the EZH2 enzyme. This, in turn, leads to the induction of angiogenesis and the growth of cancer cells [229,367]. The mechanism of the regulation of angiogenesis by miRNAs affecting histone-modifying enzymes is complex and plays a key role in the development of cancer.

## 5. Therapeutic Potential of MicroRNA-Histone Pathways

Recent advancements in cancer therapy have brought forth promising approaches based on epigenetic modifications and gene regulation. Two of these approaches are epi-drugs, focusing on histone modifications, and replacement therapy, utilizing microRNA molecules. Epi-drugs, or drugs targeting histone modifications, offer the potential for a precise regulation of epigenetic changes in histones that influence the activity of genes associated with the oncogenic process [60,368]. An example of such a drug is SAHA (suberanilohydroxamic acid), an inhibitor of histone deacetylases (HDACs), which has been approved for the treatment of myelodysplastic syndromes [369]. Other HDAC inhibitors, such as vorinostat and romidepsin, are also being explored as potential anti-cancer agents [370,371,372,373]. Additionally, inhibitors of histone methyltransferases, like DZNep, and histone demethylase inhibitors are under investigation for the treatment of cancers exhibiting aberrant histone modifications [374,375,376]. Replacement therapy using microRNA molecules focuses on regulating gene expression by manipulating the levels of microRNAs within cancer cells. AntagomiRs, which are antisense oligonucleotides, block excess microRNAs associated with cancer development. On the other hand, microRNA mimics deliver missing microRNAs that can inhibit oncogene activity or restore tumor suppressor functions [377,378,379]. Identification of specific miRNAs that interact with histone-modifying enzymes may lead to the design of more precise targeted therapies. The use of antagomirs or miRNA analogues may enable selective modulation of cancer-related gene expression [380,381,382]. This approach has the potential to impede tumor growth, but challenges related to effective microRNA delivery to target cells and issues regarding the safety and efficacy of this therapy remain subjects of ongoing research.

Preclinical research and preliminary clinical trials suggest that precisely influencing these pathways may be a promising way to inhibit tumor growth and increase the effectiveness of anticancer therapy. An important issue is also the introduction of therapeutic combinations that include miRNA inhibitors and inhibitors of histone-modifying enzymes, which can bring revolutionary effects in cancer treatment [117,313,382,383]. By implementing this strategy, scientists try to use the synergistic effect of both types of inhibitors to achieve stronger and more effective therapeutic effects. In a study by Amodio et al. [58], it was shown that miR-29b, a known tumor suppressor, specifically targets the histone deacetylase HDAC4, and both molecules are involved in a functional regulatory loop. Silencing HDAC4 using shRNA induced the inhibition of multiple myeloma cell survival and migration while stimulating apoptosis and autophagy. This also resulted in the upregulation of miR-29b through promoter hyperacetylation, which resulted in the downregulation of anticancer proteins such as SP1 and MCL-1, which were targets of miR-29b. Treatment with the pan-HDAC inhibitor, SAHA (vorinostat), also affected miR-29b expression, overcoming the negative effect of HDAC4. In vivo studies have observed strong synergism between synthetic miR-29b eccentrics and SAHA in a murine xenograft model of multiple myeloma, suggesting the therapeutic potential of this combination and representing a novel strategy to modulate the epigenome in the treatment of this cancer [58]. Brest et al. [384] found that miR-129-5p plays a key role in the action of histone deacetylase inhibitors (HDACi), such as trichostatin A and vorinostat, in the context of the anticancer activity of these substances. In their study, they used various cell lines, including papillary thyroid cancer (PTC) cells, and showed that HDACi overexpressed miR-129-5p, activated histone actylation, and induced cell death. Of particular importance was the discovery that miR-129-5p can induce cell death on its own, and its presence is necessary for the cell-killing effect of HDACi. Researchers have also shown that miR-129-5p enhances the anticancer effects of other drugs, such as etoposide, suggesting its importance in anticancer therapy. MiR-129-5p (as well as its antagomiR) has emerged as a key element in the mechanism of action of anticancer HDACi [384]. Xue et al. [385] conducted a study to explore the mechanisms behind the development of resistance to histone deacetylase inhibitors in nasopharyngeal cancer (NPC). They established SAHA-tolerant NPC cell lines that exhibited reduced apoptosis when treated with SAHA. The study revealed that the downregulation of miR-129 played a significant role in SAHA tolerance, and manipulating miR-129 expression overcame this tolerance, enhancing SAHA-induced apoptosis. Furthermore, the researchers identified NEAT1 (LncRNA) as a regulator of miR-129, showing that NEAT1 was upregulated in SAHA-tolerant cells and contributed to SAHA tolerance by modulating the miR-129/Bcl-2 axis [385].

Discovering optimal combinations of miRNA inhibitors and histone-modifying enzymes is currently being intensively researched. The selection of appropriate molecular targets that interact synergistically to inhibit carcinogenesis processes is crucial to the success of this strategy. In vitro, in vivo, and clinical studies are necessary to assess the effectiveness and possible side effects of such therapies.

## 6. Challenges and Future Directions

Despite the promising potential of miRNA–histone interactions in cancer therapy, several technological challenges persist in studying these complex regulatory mechanisms. A key challenge in developing new protocols using epi-drugs and replacement therapy is the identification of dominant microRNAs in cancer therapy. Addressing this challenge is complicated due to the diversity and complexity of microRNA regulatory mechanisms. One approach involves analyzing microRNA expression in clinical and experimental samples, as well as predicting microRNA targets and their functions using bioinformatics tools. Studying the impact of specific microRNAs on epigenetic mechanisms and their role in a particular disease can help identify dominant microRNAs that may potentially serve as therapeutic targets. Another challenge is accurately identifying and characterizing specific miRNA–histone interactions within the intricate network of gene regulation. High-throughput sequencing methods have advanced our understandings, but they often lack the resolution to capture subtle interactions. Another obstacle is deciphering the context-specific nature of miRNA–histone interactions. The effects of miRNA on histone modifications can vary depending on the cellular context and tumor microenvironment. Therefore, developing technologies that allow the assessment of miRNA-histone interactions in vivo and in relevant tissue contexts will be crucial for a comprehensive understanding. Despite these challenges, the translational prospects of harnessing miRNA–histone interactions for therapeutic interventions are highly promising. As our understanding of the underlying mechanisms improves, innovative therapeutic strategies can be developed.

It is important to emphasize that microRNA therapy and epigenetics in the context of cancer treatment are areas of intensive research, and further research efforts are necessary to gain a more precise understanding of the potential of these therapeutic strategies. Nevertheless, they hold promise for offering new possibilities for more effective and personalized anticancer therapies. Precision medicine approaches hold significant potential. By profiling miRNA expression patterns and histone modification signatures in individual patients, experiments can tailor treatment strategies based on the unique molecular characteristics of each tumor. Advances in genome editing technologies like CRISPR-Cas9 offer the possibility of directly manipulating miRNA or histone enzyme levels, potentially restoring normal regulatory pathways disrupted by cancer. Nanotechnology and delivery systems also provide avenues for overcoming challenges in targeted modulation. Developing nanoparticle-based systems to deliver miRNA mimics or inhibitors, as well as small molecules targeting histone-modifying enzymes, could enhance the specificity and effectiveness of therapeutic interventions. Furthermore, multi-omics approaches integrating miRNA, epigenetic, and transcriptomic data will likely unveil intricate networks of miRNA–histone interactions. This holistic view will facilitate the identification of key nodes in the network, which can then be targeted for therapeutic purposes.

## 7. Summary

This paper analyzes the important role of miRNA interactions with histone-modifying enzymes in the context of cancer. The studies presented in the text show the complex mechanisms by which miRNA affects the regulation of the activity of histone enzymes, influencing the expression of genes related to the proliferation, differentiation, and apoptosis of cancer cells. This understanding of the mechanisms of these interactions opens the door to potentially innovative anti-cancer therapies that aim to restore the proper regulation of genes crucial for carcinogenesis. The future of research in the area of the interaction of miRNAs with histone-modifying enzymes in the context of cancer therapy seems extremely promising. Increasing knowledge of specific miRNAs and their role in the regulation of histone enzymes may open new doors for the design of more targeted and personalized cancer therapies. The combination of advanced RNA sequencing, bioinformatics and gene therapy technologies could enable the targeting of specific miRNA-histone interactions, which could ultimately lead to revolutionary breakthroughs in the treatment of various types of cancer. However, further study of these mechanisms, both in laboratories and in clinical trials, is necessary to gain a full understanding and confirmation of the effectiveness of these therapies. As new miRNAs and their role in tumorigenesis are discovered, therapy based on miRNA-histone interactions could become a key tool in the fight against cancer and open new horizons in the field of oncology.

All figures were created using BioRender software (https://app.biorender.com/, accessed on 25 August 2023).

## Figures and Tables

**Figure 1 biomolecules-13-01590-f001:**
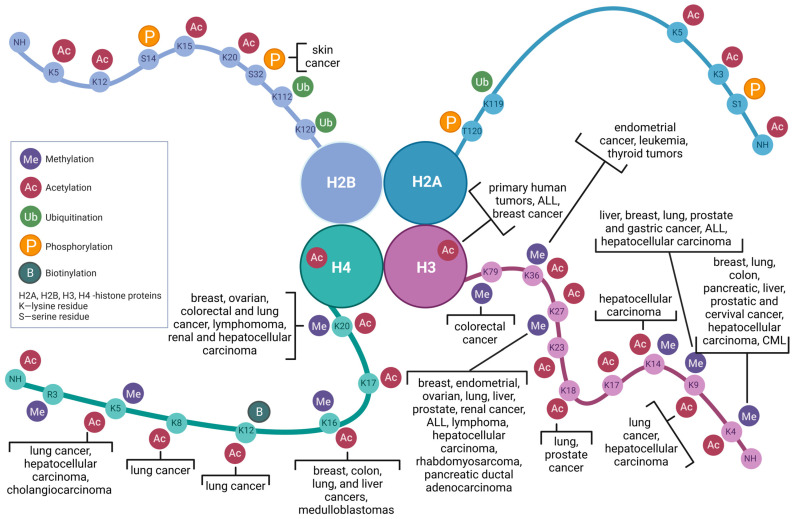
Correlation between Histone Modifications and Distinct Neoplastic Entities.

**Figure 2 biomolecules-13-01590-f002:**
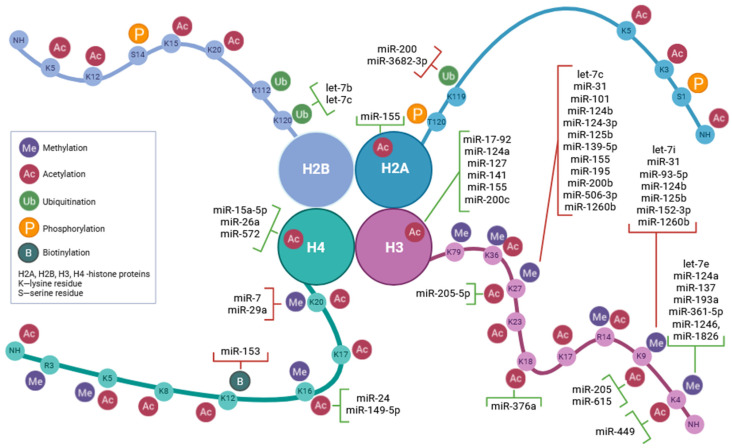
Regulation of Epigenetic Modifications of Histones by miRNA. Regions marked in red indicate inhibition, while regions highlighted in green indicate activation of histone processes.

**Table 1 biomolecules-13-01590-t001:** Selected MicroRNAs and Their Roles in Oncogenesis and Tumor Progression.

MicroRNA	Cancer Type	Role of miRNA	Target Gene(s)	Ref.
**Tumor Suppressor microRNA** TSmiRs-underexpressed in cancer cells; inhibit cancer development by negatively suppressing the function of oncogenes and/or mRNAs that modulate the cell proliferation and cycle				
let-7 family	Breast, nasopharyngeal, and oral cancer	regulates EMT process	*RAS* family, *HMGA2*, *MYC*, *OSM*	[13,14,15,16,17,18,19,20,21]
miR-15/16	Melanoma, bladder, colorectal, and prostate cancer, pituitary adenomas, CLL	induces apoptosis and inhibits cancer progression; contributes aggressiveness, drug resistance	*BCL2*, cyclin *D1*, *MCL1*, *CDC2*, *ETS1*, *JUN*, *ROR1*	[22,23,24,25]
miR-140	Colorectal, lung, breast, and ovarian cancer	inhibits cancer progression and liver metastasis, promotes apoptosis	*BCL9*, *BCL2*, *PDGFRA*, *ATP8A1*, *IGF1R*	[26,27,28,29,30]
miR-148a	Bladder, gastric, colorectal, pancreatic, and non-small cell lung cancer	regulates growth and promotes apoptosis	*BCL2*, *DNMT1*, *CCK*-*BR*	[31,32,33,34,35]
miR-340	Colon, ovarian, gastric, and endometrial cancer	triggers apoptosis and inhibits cell proliferation, repression of Wnt pathway	*Notch*, *BCL2*, *BIM*, *Bax*, *RLIP76*, *REV3L*, pro-caspase 3, β-catenin, *LGR5*, *FHL2*, *NF*-*κB1*	[36,37,38,39,40]
miR-34 family	NSCLC, AML, prostate, colorecta, breast, and lung cancer	increases apoptosis, inhibits oncogene expression, represses proliferation, cell cycle progression, and induces apoptosis	*SYT1*, *PDL*-*1*, *CDK4*	[36,41,42,43,44,45,46,47]
miR-200 family	Breast, thyroid, bladder, and prostate cancer	regulates EMT mechanism	*TGFβ*, *ZEB1*, *ZEB2*, *SNAIL*, *TWIST*, *β*-catenine, *PDGF*-*D*	[48]
miR-19	Gastric cancer	inhibits cell proliferation, repression of Wnt pathway	*MEF2D*	[49]
miR-133a	Gastric, colorectal, cervical, and pancreatic cancer	suppresses proliferation	*TCF*, *IGFF1R*, *EGFR*, *FSCN1*	[50,51,52,53,54]
miR-29	Lymphoma, retinoblastoma, leukemia, melanoma, cervical, and lung cancer	regulate multiple oncogenic processes, including epigenetics, proteostasis, metabolism, proliferation, apoptosis, metastasis, fibrosis, angiogenesis, and immunomodulation	*CDK6*, *DNMT3B*, *TCL1*, and *MCL1*	[55,56,57,58]
**Oncogenic microRNA** OncoMiRs-overexpressed in cancer cells; promote cancer development and progression by downregulating expression level and function of tumor suppressor gene				
miR-125	Pancreatic, colorectal, thyroid and gastric cancer, NSCLC	Stimulates proliferation and invasion	*EphA2, TAZ, TEAD2, TRIAP1, FNDC3B*	[59,60,61,62,63]
miR-103	Bladder, colorectal, breast and pancreatic cancer	Promotes tumor development and metastasis	*DAPK, KLF4, PTEN*	[64,65,66,67]
miR-107	Pancreatic and cervical cancer, osteosarcoma	Stimulates proliferation and migration	*SALL4, FEZF1-AS1*	[68,69,70]
miR-17-92 cluster (including miR106a, miR17-5p, miR19a, miR25, miR93)	Lung, thyroid, breast, colon and bladder cancer	Promotes tumor development through apoptosis inhibition	*ZBTB4, BCL2L1 MYCN, GAB1, RBL1, TSG101, p63, STAT3*	[71,72,73,74,75,76]
miR-21	Cervical, colorectal cancer, NSCLC, CML	Promotes proliferation, invasion, and metastasis; Inhibits apoptosis, regulates cell cycle; Increases tumor aggressiveness	*PTEN, BCL11B, Ras, KRIT1*	[77,78,79,80]
miR-155	Hepatocellular cancer, osteosarcoma, brain cancer	Enhances inflammatory response, promotes angiogenesis; Controls proliferation, regulates apoptosis; Augments angiogenesis and invasion	*TP53, PI3K, SHIP1*	[81,82,83]
miR-106b-25 (including miR-106b, miR-93, and miR-25)	Breast, prostate, lung cancer, gastric, colorectal, hepatocellular and esophageal cancer	controls cell proliferation, migration, invasion, and metastases	*LARP4B, DAB2, REST-1, ALEX1, FUT6, RUNX3*	[84,85]
Dual role microRNA				
miR-221/222	Breast, colorectal and epithelial cancers, myeloma, glioma	Enhances cancer cell survival; Inhibits angiogenesis, reduces proliferation; Controls angiogenesis and tumor cell growth	*PUMA, TRPS1, PTEN*	[86,87,88,89,90,91,92,93,94,95]
miR-146a	Prostate, breast, and colon cancer, NSCLC	Modulates inflammation, angiogenesis; Suppresses tumor growth and progression; Regulates tumor microenvironment	*Rac1, Notch2, TNFalpha, SOX5, TRAF*	[96,97,98,99,100,101,102]

**Table 2 biomolecules-13-01590-t002:** Histone Modifications and Their Impact on Gene Expression.

Histone Modification Type	Consequences on Gene Expression in Cancer	Examples in Cancer	Ref.
Acetylation (e.g., H3K9ac, H3K27ac)	Enhances transcription: loosens chromatin structure, promoting accessibility of transcription machinery and co-activators.	Overexpression of p300/CBP acetyltransferases in prostate cancer promotes H3K27 acetylation, enhancing androgen receptor-mediated transcription.	[136,137,138]
Methylation (e.g., H3K4me3)	Activation marks: enrichment at gene promoters is associated with active transcription initiation.	H3K4me3 marks are found at promoters of genes involved in cell cycle regulation, such as *MYC* and *CCND1*, in breast cancer.	[119,139,140,141]
Methylation (e.g., H3K9me2/3, H4K20me3, H3K27me3)	Silencing marks: dense methylation at certain sites is linked to gene repression, impacting chromatin compaction.	H3K27me3-mediated silencing of tumor suppressor genes, such as *CDKN2A*, is observed in various cancer types.	[112,119,141,142,143,144]
Methylation (e.g., H3K4me3, H3K36me3)	Transcription activation: associated with active transcription and splicing.	H3K36me3 is enriched in the bodies of actively transcribed genes, including *PTEN* and *TP53*, in renal cell carcinoma and glioblastoma.	[112,144,145,146]
Phosphorylation (e.g., H3S10ph)	Transcription activation: occurs during gene activation, aids chromatin decondensation and transcription factor recruitment.	Phosphorylation of H3S10 is associated with upregulation of proto-oncogenes, such as *MYC*, in leukemia.	[147]
Ubiquitination (e.g., H2BK120ub)	Transcription regulation: affects gene expression via multiple mechanisms, including recruiting transcriptional regulators.	H2BK120ub facilitates recruitment of DNA damage repair factors at the *BRCA1* gene locus in breast cancer cells.	[148,149]
SUMOylation (e.g., H4K12su)	Transcriptional repression: can lead to heterochromatin formation, silencing gene expression.	Sumoylation occurs on both oncogenes such as *MYC* and β-catenin and tumor suppressors such as p53, *PTEN*, and *BRCA1*	[150,151]
ADP-ribosylation (e.g., H2B-Glu35)	Gene silencing: impedes access to DNA, contributing to chromatin condensation and gene repression.	PARP-mediated ADP-ribosylation of histone H1 is linked to DNA repair at the *BRCA1* gene promoter in breast cancer cells.	[152,153]
Crotonylation (e.g., H3K9cr, H3K122cr)	Transcription activation: associated with actively transcribed genes, potentially enhancing transcription.	Increased interferon activation and inhibition of the tumorigenic potential of glioblastoma stem cells, leading to enhanced infiltration of CD8+ T cells and slowed tumor growth.	[154,155]

**Table 3 biomolecules-13-01590-t003:** Diverse Classes of Histone-Modifying Enzymes and Their Crucial Roles in Shaping Chromatin Modifications and Gene Expression.

Enzyme Category	Enzyme Types	Substrate Histones	Type of Modification	Ref.
Histone Acetyltransferases (HATs)	p300/CBP, GCN5, PCAF, TIP60, hCLOCK	H3, H4	Acetylation (e.g., H3K9, H3K14, H4K5)	[156,157,158,159]
Histone Lysine Methyltransferases (KMTs)	SET1, SET8 SUV39H1, SUV39H2, EZH2, ASH1L, NSD1, SMYD3, DOT1L	H3, H4	Methylation (e.g., H3K4, H4K5, H3K9, H4K20, H3K27, H3K36, H3K79)	[160,161,162,163]
Histone Arginine Methyltransferases (RMTs)	PRMT1,4,5,9	H3, H4	Methylation (e.g., H3R8 and H4R3)	[164,165,166,167]
Histone Phosphorylating Kinases	MSK1, CDKs	H3	Phosphorylation (e.g., H3S10)	[147,168,169]
Histone Ubiquitin Ligases	RNF20/40, BRCA1	H2B	Ubiquitination (e.g., H2BK120)	[170,171,172,173,174]
Histone Deacetylases (HDACs)	(1) Class I Rpd3-Like Proteins (HDAC1, HDAC2, HDAC3, and HDAC8) (2) Class II Hda1-Like Proteins (HDAC4, HDAC5, HDAC6, HDAC7, and HDAC9) (3) Class III Sir2-Like Proteins (SIRT1, SIRT2, SIRT3, SIRT5, SIRT6, and SIRT7)	H3, H4	Deacetylation (e.g., H3K9, H3K27)	[124,126,175,176,177,178]
Histone Demethylases	(1) Histone lysine demethylases KDM1-8 families (e.g., KDM1A (LSD1), KDM2B (FBXL10), KDM3A (JMJD1A), KDM4B (JMJD2B), KDM5A-D (JARID1A-D), KDM6A (UTX), KDM7 (PHF2), (2) Arginine Demethylase (JMJD6)	H3	Demethylation (e.g., H3K4, H3K27)	[124,126,129,141,175,176,177,178,179,180,181]
Histone Kinases	MSK1,2, PKC, RSK2, JAK2, MAP3K8, LIMK2, NEK6, BUB1, CHEK2, PAK2	H3, H2A	Phosphorylation (e.g., H3S10, H2AS1)	[147,182,183,184]
Ubiquitinating Enzymes	RBX1, RNF8, HUWE1, and UHRF1	H2A, H2B	Ubiquitinations (e.g., H2BK120ub)	[149,185,186,187]
Deubiquitination Enzymes	USP3, USP7, USP11 and USP22	H2A, H2B	Deubiquitination (e.g., H2AK119, H2BK120)	[188,189,190]
Poly(ADP-ribose) Polymerases (PARPs)	PARP1	H1	ADP-ribosylation (e.g., H1)	[190,191,192]
SUMO Ligases	PIAS1, PIAS4	H2B	SUMOylation (e.g., H2BK126)	[193,194,195]
Desumoylation Proteins	SENP1	H2B	Desumoylation (e.g., H2BK126)	[196,197]
Histone Crotonyltransferases	CAT2A, CAT2B, GCN5	H3, H4	Crotonylation (e.g., H3K9)	[198,199,200]

**Table 4 biomolecules-13-01590-t004:** MicroRNA-Mediated Modulation of Histone-Modifying Enzymes and Their Implications on Chromatin Modifications and Gene Expression.

MicroRNA	Targeted Histone-Modifying Enzymes	Consequences of Interaction	Impact on Gene Expression	Ref.
miR-26a, miR-101, miR-1	EZH2	Suppression of EZH2: Reduces trimethylation of H3K27, leading to derepression of silenced genes.	Derepression of target genes, influencing cell differentiation.	[269,280,281,282,283,284,285]
miR-200 and miR-221/222 families, miR-206	SUZ12, BMI1	Suppression of SUZ12/BMI1: Impedes PRC2 complex function, affecting H3K27me3 marks.	Altered chromatin states and changes in gene expression profiles; modulation of genes associated with cellular differentiation.	[286,287]
miR-214	EZH2	Suppression of EZH2: Reduces H3K27me3 levels, influencing chromatin accessibility.	Activation of genes associated with tumor suppression.	[286,287,288,289,290]
miR-101, miR-188, miR-211, miR-30d	HDAC9	Suppression of HDAC9: Disrupts deacetylation, leading to altered chromatin structure.	Activation of genes associated with cell cycle regulation.	[291,292,293,294,295]
miR-449a, miR-210	HDAC1	Suppression of HDAC1: Impedes deacetylation, influencing chromatin compaction.	Impact on genes associated with cellular responses.	[296,297,298,299]
miR-22, miR- 27b, miR-206, miR-221, miR-433	HDAC6	Suppression of HDAC6: Alters acetylation balance, influencing gene expression.	Modulation of genes associated with cellular processes.	[300,301,302,303,304]
miR-16, miR-15b, miR-200 family	SUZ12	Suppression of SUZ12: Impedes PRC2 activity, affecting H3K27me3 marks.	Altered gene expression patterns and cellular responses.	[287,305,306]
miR-203	BMI1	Suppression of BMI1: Disrupts PRC1 activity, affecting H2AK119ub marks.	Changes in gene expression profiles and cellular functions.	[285,307,308,309,310,311,312]

## Data Availability

Not applicable.

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
