# Peer review of "MicroRNA-Mediated Regulation of Histone-Modifying Enzymes in Cancer: Mechanisms and Therapeutic Implications"

_biomolecules, 2023, doi:10.3390/biom13111590_

Round 1

Reviewer 1 Report

Comments and Suggestions for Authors

In this manuscript, the authors discussed and summarized the relationship between miRNAs and histone-modifying enzymes in the context of cancer. Through the direct or indirect regulation mechanism, miRNAs modulated the expressions of histone-modifying enzymes which further influence the carcinogenesis process including cell proliferation, differentiation, apoptosis and angiogenesis. 

Below are the points to address:

Table 1 lists the selected miRNA exhibiting dual roles in oncogenesis and tumour progression. However, the statements require modifications, e.g., in Table 1, “miR-34a’s oncogenic function is to induce apoptosis and halt cell cycle”, and “miRNA-200 family members’ oncogenic function are EMT inhibitors”. miRNA-34a and miRNA200 family are known to be TS miRs. MiR-34a overexpression has been used in clinical trials as a potential treatment for cancers. Please separate TSmir, Oncomir and dual-role miRNAs in the table.

Line 120 Methylation activates or represses gene expression depending on which residue is methylated. Please revise the sentence below “Histone acetylation and methylation are often associated with gene activation, including oncogenes that drive tumorigenesis [56,142].”

Line 169-173 The authors enumerated several miRNAs that directly interact with proteins regulating the epigenetic state by histone modifications. Please give examples detailing which proteins have been affected by listed miRNAs, together with references. Should these miRNAs be listed in Table 4?

In Table 4, the authors showed the miRNA-mediated modulation of histone-modifying enzymes and their implications on Chromatin modifications and gene expression. However, it was not comprehensively summarized, e.g. Line 418-420 “MiR-125b inhibits EZH2 expression, which leads to decreased histone H3K27 methylation in the regulatory regions of apoptotic genes” This information is not included in Table 4 along with miR-26a, miR-101 and miR-1. The miRNAs that modulate HDAC3/4/5 are also not listed in Table 4. Please revise Table 4 with all miRNAs and histone-modifying enzymes mentioned in the manuscript. Meanwhile, please check the reference's accuracy, e.g., the functions of miR-214 are studied in references 225-227 instead of 223-227.

Besides SAHA, are there any other inhibitors of histone-modifying enzymes known that could therapeutically combine with antagomirs for potential cancer treatment? As multiple miRNAs all possess the abilities to regulate certain histone-modifying enzymes via transcriptional repression or mRNA destabilization, how to identify the dominant miRNA among them which could benefit further cancer treatment?

Author Response

In this manuscript, the authors discussed and summarized the relationship between miRNAs and histone-modifying enzymes in the context of cancer. Through the direct or indirect regulation mechanism, miRNAs modulated the expressions of histone-modifying enzymes which further influence the carcinogenesis process including cell proliferation, differentiation, apoptosis and angiogenesis.

Dear Reviewer,

We would like to express our gratitude for your valuable insights and constructive review of our manuscript. Your suggestions have significantly contributed to improving our research work.

Below, we present a list of changes made to the manuscript in accordance with your recommendations:

Table 1 lists the selected miRNA exhibiting dual roles in oncogenesis and tumour progression. However, the statements require modifications, e.g., in Table 1, “miR-34a’s oncogenic function is to induce apoptosis and halt cell cycle”, and “miRNA-200 family members’ oncogenic function are EMT inhibitors”. miRNA-34a and miRNA200 family are known to be TS miRs. MiR-34a overexpression has been used in clinical trials as a potential treatment for cancers. Please separate TSmir, Oncomir and dual-role miRNAs in the table.

Thank you for your suggestions regarding the table, which have led to the creation of a new version with the appropriate classification of microRNAs into tumor suppressor miRs (TSmir) and OncoMiRs in the context of oncogenesis and tumor progression. We would like to further explain that the purpose of this table was to provide a general overview of the diverse roles of microRNAs in cancer biology. In the table, we aim to encompass various functions of miRNAs, and we are aware that classifying miRNAs as TSmir or OncoMiRs can be more intricate in certain cases and context-dependent. As scientific research progresses and new evidence emerges, this classification may evolve. Therefore, the table represents the current state of knowledge and is subject to adjustment as new information becomes available.

Line 120 Methylation activates or represses gene expression depending on which residue is methylated. Please revise the sentence below “Histone acetylation and methylation are often associated with gene activation, including oncogenes that drive tumorigenesis [56,142].”

Thank you for your valuable feedback. We have revised the sentence as per your suggestion to better describe that histone acetylation and methylation influence gene regulation, both activation and repression. This more accurately reflects the role of these processes in the context of oncogenes and tumorigenesis.

Line 169-173 The authors enumerated several miRNAs that directly interact with proteins regulating the epigenetic state by histone modifications. Please give examples detailing which proteins have been affected by listed miRNAs, together with references. Should these miRNAs be listed in Table 4?

Thank you for the suggestion regarding specific examples of miRNAs directly interacting with proteins that regulate the epigenetic state through histone modifications. In response to this suggestion, we have included a few selected examples aimed at illustrating the diversity of possibilities in this phenomenon. We have presented these examples along with information about specific target genes and the cancers in which specific modifications have been described, along with references to the sources.

Please note that these examples are merely selected illustrations of the multitude of possibilities for miRNAs in epigenetic regulation and their impact on proteins. There are many other miRNAs, target proteins, and cancers that are also under investigation in this context. In the subsequent sections of the paper, we provide more detailed descriptions of some of these examples or present them in greater detail in the aforementioned Table 4. Due to the scope of our paper, we are unable to add and discuss too many examples in depth. Our focus is on presenting the direction of research and its potential. However, if you believe it is essential, we are certainly open to significantly expanding the scope of the description or Table 4.

In Table 4, the authors showed the miRNA-mediated modulation of histone-modifying enzymes and their implications on Chromatin modifications and gene expression. However, it was not comprehensively summarized, e.g. Line 418-420 “MiR-125b inhibits EZH2 expression, which leads to decreased histone H3K27 methylation in the regulatory regions of apoptotic genes” This information is not included in Table 4 along with miR-26a, miR-101 and miR-1. The miRNAs that modulate HDAC3/4/5 are also not listed in Table 4. Please revise Table 4 with all miRNAs and histone-modifying enzymes mentioned in the manuscript.

Thank you for the suggestion regarding Table 4 and the feedback concerning the absence of some information in the table. We would like to emphasize that our goal was to provide readers with selected examples of miRNAs and their influence on histone modifications in a way that allows for quick reference in the text. Due to the limited scope of the table and the desire to avoid excessive repetition of information from the text, we do not intend to significantly expand Table 4 to include all miRNAs and histone-modifying enzymes discussed in the manuscript.

However, we are open to other suggestions for improving the table or adding other elements that facilitate navigation for readers if you deem it necessary.

Meanwhile, please check the reference's accuracy, e.g., the functions of miR-214 are studied in references 225-227 instead of 223-227.

Thank you for bringing the issue with the references to our attention. We appreciate your diligence in ensuring the accuracy of the citations. We reviewed and corrected the reference numbering in our manuscript to accurately reflect the source of information regarding the functions of miR-214. We apologized for any confusion this may have caused and made the necessary corrections in our document. Your feedback was valuable in ensuring the quality and accuracy of our manuscript..

Besides SAHA, are there any other inhibitors of histone-modifying enzymes known that could therapeutically combine with antagomirs for potential cancer treatment? As multiple miRNAs all possess the abilities to regulate certain histone-modifying enzymes via transcriptional repression or mRNA destabilization, how to identify the dominant miRNA among them which could benefit further cancer treatment?

You posed important questions regarding the potential therapeutic use of inhibitors of histone-modifying enzymes in combination with antagomiRs for cancer treatment and the method of identifying dominant miRNAs in this context.

Regarding the first question, there are several other inhibitors of histone-modifying enzymes being explored for potential cancer therapy. Examples include HDAC inhibitors other than SAHA, such as vorinostat or romidepsin, as well as inhibitors of histone methyltransferases like DZNep. Combinations of these inhibitors with antagomiRs may represent an interesting therapeutic strategy but require further research and clinical testing.

As for the second question, identifying the dominant miRNA in the context of cancer therapy can be challenging due to the diversity and complexity of miRNA regulation mechanisms. One approach is to analyze the expression of miRNA in clinical and experimental samples, along with predicting miRNA targets and their functions using bioinformatics tools. Research into the impact of specific miRNAs on epigenetic mechanisms and their role in a particular disease can help identify dominant miRNAs that may potentially be therapeutic targets.

However, it's important to note that miRNA therapy and epigenetics in the context of cancer treatment are areas of intense research, and further research work is necessary to gain a precise understanding of the potential of these therapeutic strategies.

We have added a brief comment on this matter.

-------

We appreciate your professionalism and dedication in the review process. Your input has been invaluable in refining our manuscript. We hope that the implemented changes meet your expectations and make our work more valuable and coherent.

With respect, Joanna Szczepanek

Reviewer 2 Report

Comments and Suggestions for Authors

The authors of the review manuscript entitled "MicroRNA-Mediated Regulation of Histone-Modifying Enzymes in Cancer: Mechanisms and Therapeutic Implications" Joanna Szczepanek and  Andrzej Tretyn have done a commendable job comprehensively accounting for the role of miRNAs and its interaction with histone modifying enzymes in the context of cancer. Additionally the authors capture the mechanisms and factors influencing gene expression in cancer cells. I strongly believe this review would serve as a preface to enthusiasts and researchers who would like to perform extensive research in miRNA field. Furthermore, this review also highlights the potential front of therapeutic cancer drugs. In particular the roles of miRNAs in regulating histone modifying enzymes could pave way for the development of personalized cancer therapies. Overall the manuscript is well written and is substantiated with informative figure panels that aids in this review article.

Author Response

Thank you very much for your positive review of our manuscript. We are pleased that our work has met with your approval and that you believe our review can be a valuable resource for enthusiasts and researchers interested in the miRNA field in the context of cancer. We appreciate your comments regarding our figures, which we tried to align as closely as possible with the content of the article to assist readers in understanding the discussed concepts. Your suggestion about the potential cancer therapies based on the role of miRNA in regulating histone-modifying enzymes is very valuable. Indeed, developing personalized cancer therapies based on these mechanisms can be an exciting area of research.

Once again, thank you for your positive feedback and constructive comments. We appreciate your support and interest in our work.

Reviewer 3 Report

Comments and Suggestions for Authors

Biomolecules

General Comments 

1.    Content  

Although it may seem that the authors have chosen a topic which could be narrowed down enough to cover a specific area, in fact it is a very wide area , with more than 2000 miRNAs to cover and a multitude of mRNAs coding for enzymes which modify histones and are altered in Cancer . It is therefore difficult to present a comprehensive  cover of the area, which they have attempted to do. The result is that the text is too dense and  tables and figures which should give some integration of the content are inadequate. Moreover they distinguish direct  effects where an mi RNA binds to the mRNA for a histone modifying  enzyme from indirect effects where the interaction is with some intermediate mRNA which  can affect the level or function of the enzyme. For this they select a few activities like transcription factors , lnc RNAs , signal pathways , but the whole of cell metabolism is available  for the miRNAs to indirectly affect the histone enzyme activites .The section on indirect effects could be reduced considerably and it should be noted how broad this area could be and less easy to target. 

2.    Relation to Cancer

Instead of focusing on the enzymes which are known to be aberrantly expressed in Cancer and in a couple of cases are targets for Cancer therapy (EZH2 and LSD1), the authors have tried to include in a general way a whole range of enzymes without reference to whether they are altered in Cancer . 

In Table 2 Why this selection of lysines for modification? There is no mention of polycomb and trithorax – (the antagonistic complexes where KMT6 (K27)and KMT2 (K4 ) methylases   are found) -   to justify focusing on K27 and K4. Methylated K36 can also induce transcription. In general, the authors seem less familiar with the enzymes than they are of the miRNAs .Table 3, with a few references aims to summarise the field of Histone modifications, with no reference to Cancer. Having a separate list of enzymes and the sites which can be acted upon is not useful . The authors have not given the correct nomenclature for histone demethylases taking methyl groups off K27 (KDM6A and B ) or K4 (KDM5 A,B,C,D and KDM1 [LSD1and 2 ]) Again lack of familiarity with the enzymes and just general activities- not particularly related to Cancer .

Table 4 does seem to relate to direct effects on enzymes seen in Cancer but it is necessary to read the references  to realise that there is a cancer connection . the Table reports on “ Chromatin modifications and Gene expression “

Specific Comments

·      Text in lines 161-164 explaining the direct function of the miRNAs should be earlier in the paper and the authors should always make sure that they are referring to the effect is on the mRNA coding for an enzyme . In the text they write as though the miRNA is interacting with the enzyme.(lines 167-169) Lines 164-166 the sentence doesn’t make senseMatter?  

·      Table 1 points out that the functions of the miRNAs can have anti and pro cancer effects and somehow in the third column summarises what the actual effect is. Are the anti and pro effects referred to in the same reference. If they are separate – indicate.  How were columns 1 and 2 integrated to give the  3rd column . NB there is no indication which of these effects are direct or indirect . 

Overall Comment

The manuscript needs to be shortened and should focus on the effects of miRNAs on histone enzymes which are well know to be altered in cancer known. The section on indirect effects should also be shortened. 

·      

Comments on the Quality of English Language

English is mainly fine. Should not referent to interaction of miRNA with and enzyme but rather mi RNA interacting with the mRNA coding fir the enzyme 

Author Response

Dear Reviewer,

we thank you for the opportunity to receive a constructive review of our manuscript on the influence of miRNA on histone-modifying enzymes in the context of cancer. Your comments are a valuable contribution to the development of our manuscript, and we would like to share our approach to your feedback and present the steps taken to improve our work. Our manuscript is a response to the growing interest in the role of miRNA in the regulation of epigenetic processes and their association with disease development, particularly cancer. As a result of your review, we have recognized the need to further narrow the scope of our work to focus more on the essential aspects of miRNA and histone-modifying enzymes related to cancer.

In the following part of our response, we will provide specific comments on your feedback and outline the action plans for improving the manuscript. Our aim is to deliver a more clear and comprehensible document that is better suited for the analysis and discussion of the influence of miRNA on histone-modifying enzymes in the context of cancer.

General Comments

  1. Content

Although it may seem that the authors have chosen a topic which could be narrowed down enough to cover a specific area, in fact it is a very wide area , with more than 2000 miRNAs to cover and a multitude of mRNAs coding for enzymes which modify histones and are altered in Cancer . It is therefore difficult to present a comprehensive  cover of the area, which they have attempted to do. The result is that the text is too dense and  tables and figures which should give some integration of the content are inadequate. Moreover they distinguish direct  effects where an mi RNA binds to the mRNA for a histone modifying  enzyme from indirect effects where the interaction is with some intermediate mRNA which  can affect the level or function of the enzyme. For this they select a few activities like transcription factors , lnc RNAs , signal pathways , but the whole of cell metabolism is available  for the miRNAs to indirectly affect the histone enzyme activites .The section on indirect effects could be reduced considerably and it should be noted how broad this area could be and less easy to target.

Your comment regarding the scope of our study is well understood, and we wish to clarify our approach to this matter. Our intention was to present the topic as comprehensively as possible, taking into consideration the diversity of miRNA and the mRNAs encoding histone-modifying enzymes associated with cancer. We aimed to provide readers with a comprehensive view of this research area.

One of the points that was raised in your review is the issue of the manuscript's length and scope. We would like to express our understanding of this matter. It is worth noting that the other reviewers did not raise any concerns or objections regarding the length and scope of our work. Therefore, in accordance with their opinion and considering the fact that there are no consistent comments on this issue, we would prefer not to significantly alter the manuscript's content in terms of length.

 However, in line with your suggestion, we acknowledge the need to adjust the presentation of the material to make it more understandable and accessible. In response to your comment, we have made modifications to certain tables and significantly revised the section regarding indirect effects. Our objective is now to deliver a more efficient and comprehensible presentation of this information, which will benefit readers and enhance their understanding of the influence of miRNA on histone-modifying enzymes in the context of cancer.  In our view, the manuscript is currently sufficiently extensive to include the necessary information and analyses that constitute a significant contribution to the field of molecular biology and the regulation of histone modification in the context of cancer. Of course, we are open to further discussion on this matter and are willing to consider specific content-related suggestions if they arise.

  1. Relation to Cancer

Instead of focusing on the enzymes which are known to be aberrantly expressed in Cancer and in a couple of cases are targets for Cancer therapy (EZH2 and LSD1), the authors have tried to include in a general way a whole range of enzymes without reference to whether they are altered in Cancer .

Thank you for your comment regarding the scope of enzymes included in our review paper on the role of microRNAs in regulating histone-modifying enzymes in the context of cancer. We would like to provide a more detailed rationale for our choice to encompass a broad range of enzymes, not only those known for their aberrant expression in cancer but also those playing a crucial role in processes related to cancer progression or with therapeutic potential. Cancers vary in their molecular and biological mechanisms. Depending on the type of cancer, different molecular pathways can come into play. Therefore, our aim was to consider a wide spectrum of enzymes to provide a comprehensive overview and accommodate the diversity of cancer types. Some enzymes, while not necessarily known for their abnormal expression in cancers, may become subjects of research and therapy in the future.

One of our objectives was to identify potential novel therapeutic targets related to epigenetic regulations. Modern cancer research focuses on identifying new therapeutic targets. Incidentally, histone-modifying enzymes that were not previously considered crucial in cancer cases may now be seen as potential therapeutic targets. Hence, we decided to include these enzymes, which might hold promise in future cancer therapy. In our review, we aimed to present the current state of knowledge but also emphasize potential areas for research and therapy, including enzymes that may be significant in the future.

Although enzymes like EZH2 and LSD1 are well-known as therapeutic targets, there are also other enzymes that could be promising targets for cancer therapy. In our paper, we provided examples of such enzymes that are still under investigation in the context of anti-cancer therapy. HDAC and HAT inhibitors, as well as drugs targeting HMTs, are the subject of numerous scientific studies and clinical trials in the context of anti-cancer therapy.

 In preparing this manuscript, we based our research on studies conducted on cancer cells and tumor samples, allowing us to consider both well-established changes in enzyme expression and those still in the research phase. We hope this explanation clarifies our choice to present a broader scope of enzymes in our manuscript.

In Table 2 Why this selection of lysines for modification? There is no mention of polycomb and trithorax – (the antagonistic complexes where KMT6 (K27)and KMT2 (K4 ) methylases   are found) -   to justify focusing on K27 and K4. Methylated K36 can also induce transcription. In general, the authors seem less familiar with the enzymes than they are of the miRNAs .

We significantly revised the table, adding a column with examples of modifications in cancer (to better emphasize the connection between modifications and cancer development, as suggested in other comments). It's important to note that the choice of presented histone modifications in the table is illustrative, and one of the criteria was their significance in cancer. It's understood that it's impossible to cover all possible modifications comprehensively. Regarding the omission of the polycomb and trithorax complexes, the goal of this table is to provide a general overview of the impact of a particular type of modification on gene regulation. As emphasized now, the selection of specific amino acids being epigenetically modified, or the addition of cancer types, serves as illustrative examples of the discussed modification type (as now explicitly highlighted).

Table 3, with a few references aims to summarise the field of Histone modifications, with no reference to Cancer. Having a separate list of enzymes and the sites which can be acted upon is not useful . The authors have not given the correct nomenclature for histone demethylases taking methyl groups off K27 (KDM6A and B ) or K4 (KDM5 A,B,C,D and KDM1 [LSD1and 2 ]) Again lack of familiarity with the enzymes and just general activities- not particularly related to Cancer .

In Table 3, we presented key enzymes involved in histone modifications. These enzymes are often discussed further in the text, and this table facilitates navigation and finding information on specific histone modification types. Adding an additional column in this table that would link these enzymes to cancer could potentially introduce confusion and disrupt its clarity. However, we would like to point out that we have also added enzymes to the table as suggested by the Reviewer.

Nevertheless, we would like to emphasize that the enzymes mentioned by the reviewer are not the only ones responsible for the demethylation of H3K4 (KDM5A-D) and H3K27 (KDM6A-B). In the literature, there are reports that demethylation of H3K4 can also be carried out by KDM1A (also known as LSD1 or AOF2 in cases such as AML, glioblastoma, prostate, liver, breast, colorectal, and lung tumors), KDM1B (also known as LSD2, AOF1, or C6orf193 in glioblastoma and breast cancer), and KDM2B (also known as CXXC2, FBL10, or FBLX10 in glioblastoma). Similarly, the demethylation of H3K27 can be performed by KDM7A (also known as JHDM1D, KDM7, or KIAA1718 in prostate cancer), KDM4C (also known as JMJD2C or JHDM3C in glioblastoma), and KDM7B (also known as PHF8 in laryngeal and hypopharyngeal squamous cell carcinoma). These enzymes were mentioned as examples in our table, highlighting the diversity of enzymes involved in histone demethylation.

Table 4 does seem to relate to direct effects on enzymes seen in Cancer but it is necessary to read the references  to realise that there is a cancer connection . the Table reports on “ Chromatin modifications and Gene expression “

We appreciate the reviewer's feedback, although it's worth noting that the comment is somewhat unclear and lacks precision, making it challenging to provide specific revisions.

 Specific Comments

  • Text in lines 161-164 explaining the direct function of the miRNAs should be earlier in the paper and the authors should always make sure that they are referring to the effect is on the mRNA coding for an enzyme . In the text they write as though the miRNA is interacting with the enzyme.(lines 167-169) Lines 164-166 the sentence doesn’t make senseMatter?

Thank you for the comment. We have made the necessary revisions to the text in accordance with the suggestions.

  • Table 1 points out that the functions of the miRNAs can have anti and pro cancer effects and somehow in the third column summarises what the actual effect is. Are the anti and pro effects referred to in the same reference. If they are separate – indicate. How were columns 1 and 2 integrated to give the  3rd column . NB there is no indication which of these effects are direct or indirect .

Table 1 has been restructured to present the information in a modified format.

 Overall Comment

The manuscript needs to be shortened and should focus on the effects of miRNAs on histone enzymes which are well know to be altered in cancer known. The section on indirect effects should also be shortened.

As mentioned earlier, we have already shortened the section on indirect effects, and we have also addressed the comments regarding the scope of the manuscript.

 We would like to again thank you for your thorough review of our manuscript, as well as for your valuable comments and feedback. Your constructive input has been extremely helpful in improving our work.

We made every effort to implement as many of the suggested revisions as possible. However, we understand that some of them may not align with our original concept for this manuscript. In cases where we were unable to incorporate a particular suggestion, we endeavored to provide explanations for our decisions.

Your assistance has been invaluable and has contributed to enhancing our work. We appreciate the time and support you have dedicated to this review process. Your involvement is highly regarded.

Round 2

Reviewer 3 Report

Comments and Suggestions for Authors

The authors have taken on board this authors original comments